# Personalized and Temporal-aware Attention for Efficient Generative Recommendation

## Abstract

Recent discoveries of scaling laws in autoregressive and large generative recommendation models have attracted considerable attention in sequential recommendation. Larger models entail increased inference latency and training costs, while most recommendation applications have stringent latency constraints. Through empirical evaluations of recent self-attention-based recommendation models, we identified two critical issues that impair model performance while increasing computational costs: inadequate modeling of temporal information and KV cache redundancy. In this paper, we propose a novel method, termed Personalized and Temporal-aware Transformer for generative recommendation (PT-Recformer), to effectively mitigate these issues. On the one hand, we introduce a novel Rotary Temporal Encoding method to effectively model extreme temporal variations inherent in user contexts. On the other hand, we propose a personalized multi-head latent attention mechanism to enhance expressive power by integrating personalized and temporal features and significantly reduce KV cache costs. Comprehensive experiments conducted on several benchmark datasets demonstrate that PT-Recformer consistently outperforms state-of-the-art baselines with only 12% of the KV cache storage required by vanilla self-attention based models. Online A/B testing on a commercial platform has validated the effectiveness of the PT-Recformer in large-scale industrial settings, resulting in a 3.13% increase in effective Cost Per Mille. Our codes are available at https://anonymous.4open.science/r/GRec-B386.

## 1 Introduction

Sequential recommendation (SR) has occupied a dominant position in contemporary commercial recommendation systems, spanning diverse domains including e-commerce Chen et al. (2019), social media Xia et al. (2023), news/video feeds Wu et al. (2022), and online advertising Zhou et al. (2018), etc. Current state-of-the-art (SOTA) sequential recommendation models Wang et al. (2015); Sun et al. (2019); Zhou et al. (2019); Kang & McAuley (2018) primarily utilize self-attention mechanisms to expressly synthesize the "user context" (historical activities of users) to predict the next action.

Recent studies Zhang et al. (2023) demonstrate that autoregressive sequential recommendation models also follow the scaling laws. The success of projects like HSTU Zhai et al. (2024); Ye et al. (2025a); Rajput et al. (2023) offers the possibility of creating large-scale recommender systems with enhanced performance. However, deploying large-scale models in real-world systems faces significant challenges: firstly, current studies predominantly rely on vanilla multi-head self-attention Kang & McAuley (2018); Xie et al. (2020); Liu et al. (2025) and overlook the essential distinctions between sequential recommendation from generative language models Ye et al. (2025b); secondly, larger models also introduce efficiency issues, such as more training resource Xie et al. (2024), increased serving latency Ye et al. (2025b) and huge memory demand Lv et al. (2024). For related work, please refer to Appendix A.

Through empirical experiments, we identified that self-attention based sequential models primarily confront the following two critical issues:

**Inadequately Modeling Temporal Information**. Convention approaches often inadequately integrate temporal and positional information, either by adding them to attention weights or by blending them with latent representations Ye et al. (2025a). Recent methods, such as HSTU Zhai et al. (2024) and Fuxi-$\alpha$ Ye et al. (2025a) introduce relative attention bias to capture the time intervals between

activities in user context. However, previous studies fail to fully leverage the temporal information inherent in the user context. Based on our investigation of three benchmark datasets, temporal intervals between adjacent items range from subsecond to multiple years. For example, in the Amazon Books Reviews dataset, the upper quartile of maximum temporal gaps exceeds 45 days, with observed extremes reaching 7 years, while the lower quartile comprises subsecond intervals (detailed statistics are provided in Appendix D). Current methods lack robust mechanisms to effectively encode these extreme temporal variations inherent in user contexts, which degrades the performance of existing sequential models.

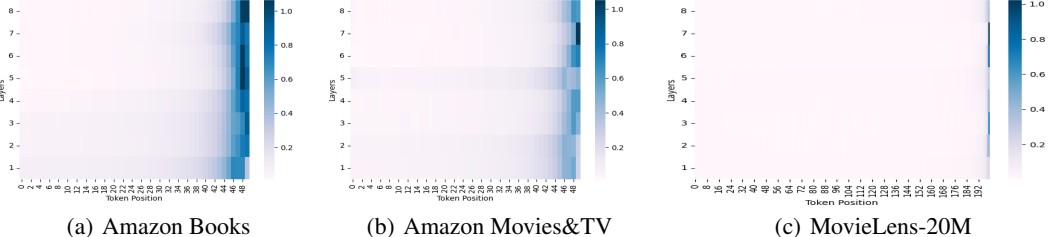

|  (a) Amazon Books | (b) Amazon Movies&TV | (c) MovieLens-20M |

Figure 1: Average attention scores generated by vanilla self-attention at each position of each layer on test datasets. The x axis is item position, and y axis is layer index. Self-attention mainly focus on latest two items in user context, resulting in huge KV cache redundancy.

**KV Cache Redundancy**. Self-attention based models in recommendation systems need to store historical Key-Value states (KV cache) for each layer and every user to save inference computational cost. In practice, a real-world commercial recommendation system may need to serve billions of users and items, resulting in prohibitive storage requirements. For example, MARM Lv et al. (2024) constructed a 60TB high-speed cache storage center, with a monthly rent that may amount to millions of US dollars[1]. However, large amounts of storage can be wasted. As illustrated in Figure 1, we compute average attention score generated by SASRec for each position in test sets of three benchmark datasets. We observed that models mainly focus on the two most recent items. This result illustrates that the self-attention based SR models may overlook remote context information, leading to substantial redundant KV cache. This result also indicates that reducing KV cache redundancy can significantly boost inference efficiency. Unfortunately, KV cache redundancy problem has received minimal attention in the recommendation academic community.

As discussed above, current self-attention based sequential recommendation models are likely to be cost-ineffective. In this paper, we propose a personalized and temporal-aware transformer architecture for sequential recommendation, named **PT-Recformer**, to address above two critical issues. Our contributions are summarized as follows:

1. We propose a novel generative recommendation model, PT-Recformer, to effectively and efficiently address two critical issues that are identified through our empirical investigation of existing approaches. To the best of our knowledge, PT-Recformer is the first personalized self-attention model tailored to the characteristics of sequential recommendation.

2. We first propose a novel rotary temporal encoding method to effectively leverage relative and large-range time intervals between interactions in the user context. Then, we propose a personalized multi-head latent attention mechanism to enhance expressive power by integrating personalized and temporal features with significantly reduced complexity.

3. Extensive experiments on public benchmark datasets consistently demonstrate that PT-Recformer outperforms SOTA baselines with significantly reduced training time and KV cache storage (achieving over 20% reduction in training time and 80% reduction in KV cache). Furthermore, we have deployed PT-Recformer on a commercial recommendation system, and online A/B testing results have validated the effectiveness of our proposed method, resulting in a 3.13% increase in eCPM.

---

[1]https://azure.microsoft.com/en-us/pricing/details/cache/

## 2  PROBLEM DESCRIPTION

We focus on the task of sequential recommendation, which is formally defined as follows: Given a user $u \in \mathcal{U}$ along with some personalized attributes $U$ (*e.g.* gender, age, etc.) and a sequence of the user's historical interactions (referred to as the context) $\{I_1, I_2, \ldots, I_n\}$ arranged in chronological order. The relative time stamps of interaction occurrence are $\{t_1, t_2, ..., t_n\}$. The objective is to recommend the next most likely item $I_{n+1}$, where $n$ is the length of user context and $I \in \mathcal{I}$. $\mathcal{I}$ represents the set of all items, *i.e.* $\forall I \in \mathcal{I}$ and $\mathcal{U}$ represents the set of all users in dataset.

$$\max_{I_{n+1} \in \mathcal{I}} P(I_{n+1} \mid I_1, I_2, \ldots, I_n, U) \tag{1}$$

Each item $I$ is associated with an ID and additional features (e.g., category, tags, text, etc.).

## 3  METHOD

The overall architecture of PT-Recformer is depicted in Figure 2. Our proposed method comprises two key components: 1) Rotary Temporal Encoding (RoTE) to represent the wide-ranging temporal interval between items in user context; 2) Personalized Multi-head Latent Attention (PMLA) to incorporate personalized information into attention with low cost.

Due to space constraints, we focus on describing the key innovations of PT-Recformer in the main body. The remaining components, such as the embedding layer, feed-forward network, and prediction layer, are detailed in Appendix B. The complexity will be analyzed in Section 4.

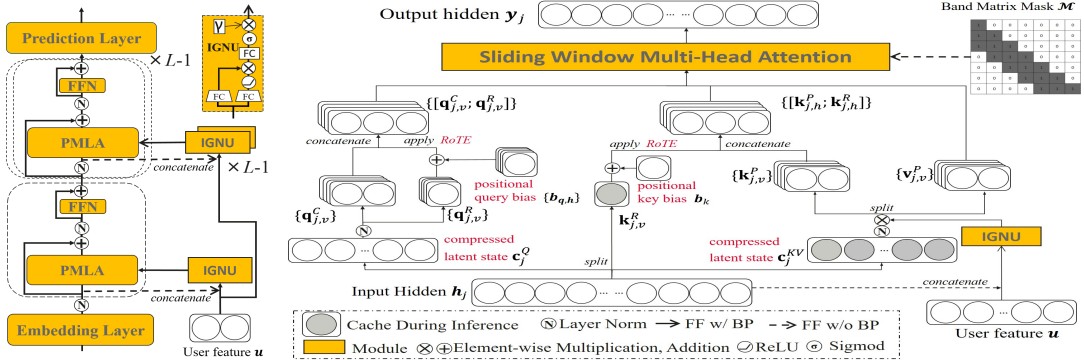

(a) Overall architecture          (b) Personalized Multi-head Latent Attention (PMLA)

Figure 2: The overall architecture of PT-Recformer and personalized multi-head latent attention. Each PMLA integrates multi-head latent attention, positional Query-Key bias and independent IGNU.

### 3.1  ROTARY TEMPORAL ENCODING (ROTE)

We propose a Rotary Temporal Encoding (RoTE) method to represent large-range time interval between items in the user context. Our proposed method contains two components: relative time-bias encoding and learnable rotary frequencies.

We first define the relative time-bias (rtb) of two items in user context. As discussed in the Introduction, the positional encoding method for sequential recommendation should account for large-range temporal intervals between items. In other words, the relative positional encoding of adjacent items should be computed differently according to their specific time intervals. Empirical observations and prior researches Zhai et al. (2024); Ye et al. (2025a) indicate that items closer to the current time should exhibit higher discrimination in positions, and vise versa. Thus, we employ logarithmic function to generate relative time-bias index. For any two items $i$ and $j$,

$$rtb_{i,j} = \min(\beta * \log(1 + |t_i - t_j|)), \max\_rbt) \tag{2}$$

where $|\cdot|$ means absolute value, $rtb_{i,j} \in [0, \max\_rbt]$. $t_i$ and $t_j$ is the timestamp of interaction time with $i$-th and $j$-th items, respectively. $\beta$ is a scalar hyperparameter and controls temporal resolution granularity. $\max\_rbt$ is the largest value of relative time-bias in the user context.

When we have a item sequence with length $n$, *i.e.* $[I_1, I_2, ..., I_n]$, the position index sequence generated by relative time-bias is defined as follow.

$$RTB = [\, rtb_{n,1}, rtb_{n,2}, ..., rtb_{n,n-1}, rtb_{n,n} \,] \tag{3}$$

The position index sequence $RTB$ is in descending order and $rtb_{n,n} = 0$. For convenience of writing, we define the j-*th* index in RTB as $RTB_j$, *i.e.* $RTB_j = rtb_{n,j}$. This design intentionally enhances positional discrimination closer to current time while reducing distinguishability for remote items.

Subsequently, we introduce a rotary encoding method with learnable frequencies for each attention head to incorporate relative time-bias into the attention mechanism. Specifically, the query and key states of the *j*-th item for each attention head, incorporating time-bias information, are generated as follows:

$$\mathbf{x}_j^R = \text{RoTE}(\mathbf{x}_j) = \mathbf{x}_j \otimes sin(RTB_j * \theta) + [-x_{d_h//2}, ..., -x_{d_h-1}, x_0, x_1, .., x_{d_h//2-1}] \otimes cos(RTB_j * \theta) \tag{4}$$

where $\mathbf{x}_j \in \mathbb{R}^{d_h}$, and $\theta \in \mathbb{R}^{d_h}$ is learnable multiple frequencies belong to a head and initialized as the frequencies in RoPE. $d_h$ is the head dimension. $sin$ and $cos$ are element-wise operation. $\otimes$ is the element-wise production.

Research in LLMs generally treats frequencies $\theta$ as hyperparameters. However, the recommendation scenarios are highly diverse, making it computationally expensive to identify the optimal frequencies for each scenario. RoTE utilizes learnable frequencies to eliminate the computational cost associated with manual frequency optimization across domains. Consequently, RoTE can be seamlessly integrated into attention-based sequential recommendation (SR) models in various scenarios with minimal additional cost. This approach introduces an additional $n_h d_h$ free parameter than RoPE Su et al. (2024), which constitutes only approximately 0.001% of the total parameters in PT-Recformer.

## 3.2 PERSONALIZED MULTI-HEAD LATENT ATTENTION (PMLA)

Conventional recommendation models often employ late fusion techniques to integrate user context and user profile features. Specifically, they first adopt a sequential model to encode variable-length user context into fixed-size hidden states. Then, they pass the hidden states along with user profile features into multilayer perceptrons to obtain the final prediction. In this paper, we contend that personalized features can also enhance user context encoding and propose a Personalized Multi-head Latent Attention (PMLA) mechanism. To the best of our knowledge, PMLA is the first personalized self-attention designed for sequential recommendation.

### 3.2.1 ATTENTION QUERY

For the attention queries, we first perform a low-rank compression with RMSNorm Zhang & Sennrich (2019) to reduce the memory usage during training.

$$\mathbf{c}_j^Q = \mathbf{W}^{DQ}\mathbf{h}_j \tag{5}$$

$\mathbf{h}_j \in \mathbb{R}^d$, and $\mathbf{c}_j^Q \in \mathbb{R}^{d_c}$ is the compressed latent vector for queries. $\mathbf{W}^{DQ} \in \mathbb{R}^{d_c \times d}$ is the down-projection matrix. $d$ is the dimension of input hidden state and $d_c \ll n_h d_h$ denotes the query compression dimensions.

The multi-head queries without temporal information is generated as follow.

$$[\mathbf{q}_{j,1}^C, ..., \mathbf{q}_{j,n_h}^C] = \mathbf{q}_j^C = \mathbf{W}^{UQ}\text{RMSNorm}(\mathbf{c}_j^Q) \tag{6}$$

$\mathbf{W}^{UQ} \in \mathbb{R}^{n_h d_h \times d_c}$ is the up-projection matrices for queries.

Then, we use our proposed RoTE and positional query bias to generate positional query states.

$$[\mathbf{q}_{j,1}^R, ..., \mathbf{q}_{j,n_h}^R] = \mathbf{q}_j^R = \text{RoTE}(\mathbf{W}^{QR}\text{RMSNorm}(\mathbf{c}_j^Q) + \mathbf{b}_q) \tag{7}$$

$\mathbf{W}^{QR} \in \mathbb{R}^{n_h d_h^R \times d_c}$ is the matrix to produce the decoupled queries that carry temporal information. $\mathbf{b}_q \in \mathbb{R}^{n_h d_h}$ is the positional query bias, and we add it only to the positional query $\mathbf{q}_j^R$ to enhance its length extrapolation ability. Subsequently, the $v$-th head query state of *j*-th item is as follow:

$$\mathbf{q}_{j,v} = [\mathbf{q}_{j,v}^C; \mathbf{q}_{j,v}^R] \tag{8}$$

where [;] means concatenate operation and $\mathbf{q}_{j,v} \in \mathbb{R}^{d_h + d_h^R}$.

### 3.2.2 INTEGRATING PERSONALIZED INFORMATION INTO THE ATTENTION KEY AND VALUE

Unlike conventional sequential recommendation models, we integrate personalized information into the attention keys and values. We also adopt low-rank compression to reduce memory usage in training too.

$$\mathbf{c}_j^{KV} = \mathbf{W}^{DKV}\mathbf{h}_j \tag{9}$$

$\mathbf{W}^{DKV} \in \mathbb{R}^{d_c \times d}$ is the down-projection matrix for keys and values.

The the personalized information is integrated into $\mathbf{c}_j^{KV}$ through our proposed Interaction Gate Neural Unit (IGNU). Given a set of user profile attributes $U$, we concatenate the hidden vectors of each features to form user feature vector $\mathbf{u} \in \mathbb{R}^{d_u}$. The input of $l$-th IGNU is the concatenation of user feature vector and the input item vector of $l$-th layer. To prevent overfitting, the gradient generated by the IGNU will not backpropagate to item hidden states.

As shown in top-right of Figure 2(a), IGNU first use two linear layer to reduces the dimension.

$$\mathbf{x}_j = \text{ReLU}(\mathbf{W}_a^1[\mathbf{u}; \mathbf{h}_j]) \otimes (\mathbf{W}_b^{(1)}[\mathbf{u}; \mathbf{h}_j]) \tag{10}$$

where $\mathbf{W}_a^{(1)}, \mathbf{W}_b^{(1)} \in \mathbb{R}^{d_c \times (d+d_u)}$. The output layer of IGNU uses sigmoid activation, and its output will be multiplied on $\mathbf{c}_j^{KV}$ element-wisely.

$$\mathbf{gate}_j = \text{Sigmoid}(\mathbf{W}^{(2)}\mathbf{x}_j) * \gamma \tag{11}$$

$$\mathbf{c}_j^{PKV} = \mathbf{c}_j^{KV} \otimes \mathbf{gate}_j \tag{12}$$

$\gamma$ is a scalar hyper parameter and limits the all elements of gate vector to $[0, \gamma]$. $\mathbf{W}^{(2)} \in \mathbb{R}^{d_c \times d_c}$. $\otimes$ is the element-wise production.

Subsequently, we use up-projection matrices to get key state without temporal information.

$$[\mathbf{k}_{j,1}^P, ..., \mathbf{k}_{j,n_h}^P] = \mathbf{k}_j^P = \mathbf{W}^{UK}\text{RMSNorm}(\mathbf{c}_j^{PKV}) \tag{13}$$

$\mathbf{W}^{UK} \in \mathbb{R}^{n_h d_h \times d_c}$ is the up-projection matrices for keys.

The positional key state is generated as follow. We do not adopt multi-head projection in positional key state and also add positional key bias $\mathbf{b}_k \in \mathbb{R}^{d_h^R}$ to enhance length extrapolation ability.

$$\mathbf{k}_j^R = \text{RoTE}(\mathbf{W}^{KR}\mathbf{h}_j + \mathbf{b}_k) \tag{14}$$

where $\mathbf{W}^{KR} \in \mathbb{R}^{d_h^R \times d}$. The $\mathbf{k}_j^R$ will be cached during inference.

The final key state of $v$-th head is generated as follow:

$$\mathbf{k}_{j,v} = [\mathbf{k}_{j,v}^P; \mathbf{k}_j^R] \tag{15}$$

Last but not least, we use up-projection matrices similar to those used for attention keys to obtain the value state.

$$[\mathbf{v}_{j,1}^P, ..., \mathbf{v}_{j,n_h}^P] = \mathbf{v}_j^P = \mathbf{W}^{UV}\text{RMSNorm}(\mathbf{c}_j^{PKV}) \tag{16}$$

$\mathbf{W}^{UV} \in \mathbb{R}^{n_h d_h \times d_c}$ is the up-projection matrices for values.

### 3.2.3 SLIDING WINDOW ATTENTION

As illustrated in Figure 1, self-attention based model may introduce significant KV cache redundancy, severely wasting computational and storage resources. Simply truncating the KV cache during inference would significantly degrade model's performance. To reduce KV cache redundancy, it is necessary to modify the attention mechanism during training. Consequently, we incorporate sliding window attention (SWA) Beltagy et al. (2020) to rectify PMLA during training. Given window size $w$, SWA restricts each item's attention to maximum of the previous $w$ neighboring items. Moreover, the receptive filed of $l$-th layer is $l \times w$, enabling deeper layers to effectively capture long-range dependencies.

We adopt a band matrix mask each layer to simplify the training procedure (top-right in Figure 2(b)). Specifically, the mask is a $n \times n$ band matrix $\mathcal{M}$ as with a bandwidth of $w$, where all non-zero entries

are set to 1. The mask will be element-wise multiplied on attention matrix in each layer. $\mathcal{M}_j$ denotes the $j$-th column of the mask. The attention output of PMLA in training is as follow.

$$\mathbf{o}_{j,v}^w = \sum_{i=1}^{n} \mathcal{M}_j \otimes \text{Softmax}(\frac{\mathbf{q}_{j,v}^T \mathbf{k}_{i,v}}{\sqrt{d_h + d_h^R}})\mathbf{v}_{i,v}^P \tag{17}$$

Therefore, we can simply truncate the KV cache to maximum of $w$ pairs per layer during inference. The incorporation of SWA substantially reduces computational and memory resource for long user context from $O(n \times d)$ to $O(w \times d)$ when $w \ll n$.

$$\mathbf{o}_{j,v}^w = \sum_{i=\max(1,j-w)}^{j} \text{Softmax}(\frac{\mathbf{q}_{j,v}^T \mathbf{k}_{i,v}}{\sqrt{d_h + d_h^R}})\mathbf{v}_{i,v}^P \tag{18}$$

The final output of our attention mechanism is as follow:

$$\mathbf{y}_j^w = \mathbf{W}^O[\mathbf{o}_{j,1}^w; ...; \mathbf{o}_{j,n_h}^w] \tag{19}$$

where $\mathbf{W}^O \in \mathbb{R}^{d \times n_h d_h}$ denotes the output projection matrix.

### 3.3 Training for Recommendation Objective

PT-Recformer follows the prediction method of autogressive generative recommendation. Thus, next item prediction is adopted and we utilize the InfoNCE loss van den Oord et al. (2018) as training objective. For any hidden state $\mathbf{h}_i$ in the output sequence of the PT-Recformer layers, the embedding of positive sample of $i$-th item is $\mathbf{e}_i^{pos} \in \mathbb{R}^d$. And the embeddings of negative samples $\mathbf{e}_i^{neg} \in \mathbb{R}^{m \times d}$ are randomly sampled from the dataset, excluding the items in current user context, and $m$ is the number of negative samples. The loss function is as follow:

$$\mathcal{L} = -\sum_{i=1}^{n} \log \frac{\exp(\mathbf{e}_i^{pos}\mathbf{h}_i^{(L)}/\tau)}{\exp(\mathbf{e}_i^{pos}\mathbf{h}_i^{(L)}/\tau) + \sum_{j=1}^{m} \exp(\mathbf{e}_{i,j}^{neg}\mathbf{h}_i^{(L)}/\tau)} \tag{20}$$

$\tau$ denotes a temperature parameter, which is a scalar hyperparameter.

## 4 Complexity Analysis

Computational efficiency plays a crucial role in recommendation systems. In Table 1, we compare several mechanisms with our PMLA and highlight the superior efficiency of our proposed approach. Multi-head Latent Interaction (MLI) is proposed by Hydra Yuan (2025), which uses multiple low-dimensional Mamba-2 blocks.

Table 1: Complexity Comparison. $n$ is context length. $d$ is dimension of embedding. $d_h$ is the head dim. $d_c$ is dimension of latent spaces. $w$ is the window size.

| | Self-Attention | Mamba-2 | MLI | MLA | PMLA |
|---|---|---|---|---|---|
| Training FLOPs | $n^2 d$ | $nd^2$ | $nvd_c^2$ | $nd_c d$ | $nd_c d$ |
| Inference FLOPs | $n^2 d$ | $nd^2$ | $nvd_c^2$ | $nd_c d$ | $wd_c d$ |
| KV Cache | $nd$ | $d^2$ | $vd_c^2$ | $n(d_c + d_h)$ | $w(d_c + d_h)$ |

1) **Training Efficiency**. Since PMLA employs low-rank compression, its the training overhead is lower than that of Self-attention and Mamba-2 and comparable to MLI. Based on our experiments, with only approximately 75% training time of SASRec, PT-Recformer can outperform SOTA baselines. 2) **KV Cache Reduction**. Due to the SWA, the KV cache cost of PMLA does not increase with the length of user context, whereas that of self-attention and MLA does. Thus, our proposed method can significantly reduce the KV cache than MLA on long user context, achieving over 80% KV cache reduction in experiments. 3) **Inference Acceleration**. Although PMLA incurs slightly higher FLOPs than standard attention for short context sequences due to up-projection on cached states, the significantly lower KV cache overhead enables our method to store much longer user contexts in the GPU HBM, substantially reducing GPU and memory I/O overhead. According to our experiments on long user context dataset (*Movielens-20M*), PT-Recformer exhibits an 18% faster inference speed than SASRec. For detailed results, please refer to the experiments in Section 5.1.1.

## 5 EXPERIMENT

To demonstrate the superiority of PT-Recformer, we first compare it with the SOTA baselines on three benchmark datasets: *Amazon Books Reviews* (Books), *Amazon Movie & TV Reviews* (Movie & TV) and *Movielens-20M*. We also conduct extensive ablation studies to reveal value of key components. Experiment details are described in Appendix C, including datasets statistics, implementation details and baselines' information, etc.

Table 2: Evaluation on Public Benchmark Datasets. Recall and NDCG values are averages over 5 random seeds and only keep 5 decimal. The best results on the same dataset are bold with $p < 0.05$, and the underline indicates the second best result.

| Dataset | Method | R@10 | R@50 | R@200 | N@10 | N@50 | N@200 |
|---|---|---|---|---|---|---|---|
| Movies & TV | HSTU-large | 8.71662 | 16.99500 | 27.69638 | 5.23617 | 7.03941 | 8.64491 |
| | SASRec | 1.50863 | 3.77692 | 9.29801 | 0.91559 | 1.39488 | 2.21743 |
| | FUXI-$\alpha$ | 7.93513 | 15.98130 | 26.45466 | 4.66556 | 6.41677 | 7.98760 |
| | Hydra | 8.65023 | 16.87432 | 27.97995 | 5.25607 | 7.29480 | 8.82005 |
| | SASRecV2 | 8.96488 | 16.88815 | 27.41993 | 5.56335 | 7.28676 | 8.86528 |
| | MLARec | 9.19535 | 17.40621 | 28.11699 | 5.65884 | 7.44542 | 9.05092 |
| | PT-Recformer | **9.23772** | **17.51174** | **28.50900** | **5.79187** | **7.52726** | **9.17153** |
| Books | HSTU-large | 6.37677 | 12.69195 | 21.20991 | 3.81513 | 5.18683 | 6.46238 |
| | SASRec | 0.76357 | 1.99900 | 4.17860 | 0.38632 | 0.64970 | 0.97549 |
| | FUXI-$\alpha$ | 6.02492 | 12.52272 | 21.3859 | 3.50377 | 4.91482 | 6.24142 |
| | Hydra | 7.03255 | 12.56186 | 20.28847 | 4.55672 | 5.75953 | 6.91643 |
| | SASRecV2 | 7.03946 | 12.99099 | 21.03707 | 4.49230 | 5.78699 | 6.99132 |
| | MLARec | 7.39836 | 13.35622 | 21.29337 | 4.74132 | 6.03856 | 7.22654 |
| | PT-Recformer | **7.56688** | **13.95876** | **22.43541** | **4.76576** | **6.15792** | **7.42781** |
| MovieLens-20M | HSTU-large | 15.53223 | 30.94855 | 48.48992 | 9.10255 | 12.48677 | 15.11581 |
| | SASRec | 1.597010 | 4.952030 | 10.44497 | 1.06936 | 1.75187 | 2.55578 |
| | FUXI-$\alpha$ | 15.03638 | 29.26782 | 46.19100 | 8.97440 | 12.06795 | 14.61086 |
| | Hydra | 16.15043 | 31.57319 | 48.07592 | 9.65599 | 13.03502 | 15.66071 |
| | SASRecV2 | 15.03638 | 28.99736 | 46.49366 | 8.83753 | 11.87361 | 14.50056 |
| | MLARec | 16.65271 | 31.06446 | 47.67210 | 9.98470 | 13.14817 | 15.65912 |
| | PT-Recformer | **17.62509** | **32.19138** | **48.52212** | **10.61433** | **13.80393** | **16.25550** |

### 5.1 MAIN RESULTS

Table 2 summarizes the overall performance across benchmark datasets. Three critical observations emerge from our analysis: 1) PT-Recformer significantly outperforms SOTA baselines at all metrics across all datasets. For example, PT-Recformer yields a consistent average improvement of approximately 15.5% and 6.7% over HSTU-large and Hydra, respectively, on *Books*. 2) Notably, PT-Recformer achieves this SOTA performance with only 25% of the KV cache required by SASRec and HSTU-large, substantially reducing computation and memory in inference. For further evaluation, please refer to Section 5.3.1 and Appendix F.1. 3) MLARec, our custom-developed baseline, achieves the second best results across most evaluation metrics. This baseline is constructed by removing IGNU and SWA from PT-Recformer. These results demonstrate not only the necessity of integrating personalized features into sequential encoding and effectiveness of our IGNU, but also the feasibility of improving SR through tailored attention. For a detailed evaluation of the impact of user profile features, please refer to the experiments in Appendix F.4 and  F.3

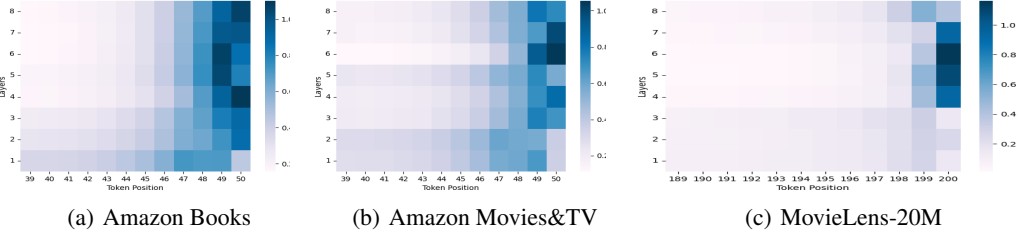

|  (a) Amazon Books | (b) Amazon Movies&TV | (c) MovieLens-20M |

Figure 3: Average attention scores of each PMLA layer on test datasets when window size is quarter of maximum user context length.The x axis is item position, and y axis is the layer index

To intuitively demonstrate the model's capacity to reduce redundant KV cache, we calculated average attention scores generated by PT-Recformer for each position in the test sets. The visualization results are presented in Figure 3. From the figure, it can be observed that, compare to vanilla multi-head attention (shown in Figure 1), the first three PT-Recformer layers all allocate some attention to remote items rather than focusing solely on most recent two items. This result provides empirical evidence supporting the reduction in KV cache redundancy achieved by our proposed method.

### 5.1.1 Training and Inference Time

To assess the time efficiency of our PT-Recformer, we compare it with SASRec and HSTU-large. Table 3 demonstrate the training and inference time on *MovieLens-20M* dataset. We trained all models from scratch five times and averaged the training and inference time across all epochs. PT-Recformer outperforms SOTA baselines with 75% training time and 18% faster inference speed. Additionaly, PT-Recformer uses only 25% memory of SASRec and HSTU for KV cache storage.

Table 3: Training and Inference Time on MovieLens-20M dataset. The time unit is second. The window size is 100, and rank size is 256.

| Method | #Param | Train time | Inference time |
|---|---|---|---|
| SASRec | 1B | 2427.62s | 499.98s |
| HSTU-large | 1B | 1970.12s | 473.64s |
| PT-Recformer | 0.37B | 1820.72s | 410.98s |

### 5.2 Ablation Study

To assess the contributions of key components in our architecture, we conducted several experiments as ablation study. Additional ablation studies are demonstrated in the Appendix.

### 5.3 Temporal Encoding

We compare our proposed RoTE with other SOTA position encoding methods on SASRecV2 and PT-Recformer, including RoPE Su et al. (2024), CAPE Yuan et al. (2025), our proposed RoTE and RoTE(fixed $\theta$),where the frequencies are frozen during training and set to the same values as in RoPE. To save cost, we only conducted experiments on *Books* dataset. The experimental results are presented in Table 4. RoTE consistently improves backbones and achieves SOTA performance, while RoTE(fixed $\theta$) also outperforms RoPE at all metrics. These results demonstrate the necessity of considering time interval between items in user context and the excellent generality of our RoTE. To the best of our knowledge, our work provides the first empirical validation of the necessity and efficacy of temporal encoding.

Table 4: Ablation study of position encoding methods on Amazon Books dataset. The best results with the same backbone are bold with $p < 0.05$.

| Method | Metrics | | | | | |
|---|---|---|---|---|---|---|
| | R@10 | R@50 | R@200 | N@10 | N@50 | N@200 |
| HSTU-large | 6.37677 | 12.69195 | 21.20991 | 3.81513 | 5.18683 | 6.46238 |
| FUXI-$\alpha$ | 6.02492 | 12.52272 | 21.3859 | 3.50377 | 4.91482 | 6.24142 |
| SASRecV2 | 7.03946 | 12.99099 | 21.03707 | 4.49230 | 5.78699 | 6.99132 |
| SASRecV2+RoPE | 7.08019 | 12.71685 | 20.43641 | 4.59212 | 5.81769 | 6.97307 |
| SASRecV2+CAPE | 7.11573 | 12.82449 | 21.56204 | 4.59985 | 5.84139 | 6.99945 |
| SASRecV2+RoTE(fixed $\theta$) | 7.16682 | 13.34342 | 21.57773 | 4.49171 | 5.83534 | 7.06873 |
| SASRecV2+RoTE | **7.17182** | **13.44752** | **21.60035** | **4.53372** | **5.89777** | **7.16543** |
| PMLA+RoPE | 7.23431 | 13.28240 | 21.53441 | 4.56499 | 5.88079 | 7.11647 |
| PMLA+CAPE | 7.23877 | 13.39911 | 21.60680 | 4.57113 | 5.91240 | 7.14130 |
| PMLA+RoTE(fixed $\theta$) | 7.52429 | 13.91027 | 22.40477 | 4.74701 | 6.13756 | 7.40596 |
| PMLA+RoTE | **7.56688** | **13.95876** | **22.43541** | **4.76576** | **6.15792** | **7.42781** |

### 5.3.1 LOW RANK COMPRESSION

We evaluate the different dimensions of of low rank compression. The embedding dimension of all models is 512. All models used 8 layers with 8 heads. The results are demonstrated in Table 5.

Table 5: Ablation study of query's and key's rank size on Amazon Books dataset. The best results on the same dataset are bold with $p < 0.05$. The window size is 12.

| $d_c$ | Metrics | | | | | |
|---|---|---|---|---|---|---|
| | R@10 | R@50 | R@200 | N@10 | N@50 | N@200 |
| 32 | 7.07486 | 12.96034 | 20.99577 | 4.53126 | 5.81031 | 7.01281 |
| 64 | 7.31993 | 13.36328 | 21.65530 | 4.67350 | 5.98949 | 7.23044 |
| 128 | 7.47953 | 13.68275 | 21.97635 | **4.74797** | 6.09863 | 7.34047 |
| 256 | **7.49363** | **13.75571** | 22.10471 | 4.73711 | **6.10066** | 7.35050 |
| 512 | 7.44060 | 13.71487 | **22.25916** | 4.70515 | 6.09896 | **7.37888** |

From the table, we can draw the following conclusions: 1) When $d_c$ is set to 32, our proposed method outperforms nearly all SOTA baselines. By adopting SWA and low-rank compression simultaneously, PT-Recformer outperforms state-of-the-art baselines using only **12%** of the memory required by SASRec for KV cache. 2) We conducted experiments on 8 GPUs with 32G memory, and PT-Recformer outperforms SOTA baselines with only **75%** training time. These results indicate that selecting an appropriate rank size can simultaneously improve model performance and reduce KV cache costs. 3) When rank size is 256, model achieves best performance at half metrics. Meanwhile, when the rank size is 512, the model does not exhibit consistent improvement despite a doubling of the KV cache cost. These results suggest that vanilla self-attention-based sequential recommendation models may lead to redundant KV cache.

### 5.4 DEPLOYMENT & ONLINE A/B TESTING

We conducted a comprehensive 7-day online A/B test within the most valuable channel of an online APP. The average length of user context is thousands. The results are shown in Table 6, providing strong evidence of PT-Recformer's effectiveness. From the table, it is immediately clear that our method consistently outperforms the baseline, resulting in a daily improvement of the north star metric, *eCPM* (Effective Cost Per Mille). This daily improvement in *eCPM* underscores the reliability and consistency of PT-Recformer's impact on performance. During the course of the experiment, PT-Recformer demonstrated an average improvement of **3.13%** relative to the highly optimized online baseline, marking a significant enhancement in the system's overall efficiency and effectiveness. Additionally, the online inference speed of PT-Recformer is **16%** faster than the same-sized Transformer-based model. These results validate the efficacy of PT-Recformer, showcasing its potential to deliver tangible improvements in online recommendation systems.

Table 6: Online results of a eight-day online A/B testing. The north star metric is eCPM.

| Day1 | Day2 | Day3 | Day4 | Day5 | Day6 | Day7 | average |
|---|---|---|---|---|---|---|---|
| +1.75% | +1.66% | +2.30% | +3.94% | +5.24% | +6.26% | +1.17% | +3.13% |

## 6 CONCLUSION

In this paper, we first identify the potential cost-ineffective risk of current self-attention based sequential recommendation models, specifically, *inadequately modeling temporal information* and *KV cache redundancy*. To address these issues, we propose a personalized and temporal-aware transformer architecture for generative recommendation, termed **PT-Recformer**, which incorporates a novel temporal encoding method RoTE and a personalized multi-head latent attention (PLMA) to leverage user profile features while reducing KV cache costs. Extensive experiments on public benchmark datasets consistently demonstrate that PT-Recformer outperforms SOTA baselines with significantly reduced training time and KV cache storage, achieving approximately 25% training time saving and over 80% KV cache storage reduction. We also conducted extensive ablation study and reveal the value of key components. Futhermore, online A/B testing on a commercial platform has validated the effectiveness of the PT-Recformer in large-scale industrial settings. To the best of our knowledge, PT-Recformer is the first personalized self-attention mechanism designed for sequential recommendation, which is anticipated to attract considerable interest from the research community.

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

# A  RELATED WORK

## A.1  ATTENTION-BASED MODELS IN SEQUENTIAL RECOMMENDATION

Recently, the remarkable success of large language models (LLMs) Yang et al. (2024); Touvron et al. (2023); OpenAI et al. (2024), exemplified by GPT OpenAI et al. (2024), has prompted many studies to introduce the Transformer architecture into sequential recommendation Wu et al. (2024), such as SASRec Kang & McAuley (2018), FUXI-$\alpha$ Ye et al. (2025a) and others Chen et al. (2019); Zhou et al. (2020); Wang et al. (2023). While self-attention mechanisms remain the foundational component of Transformer architectures, extensive architectural innovations DeepSeek-AI et al. (2025); Jiang et al. (2023); Lou et al. (2024) have been proposed in NLP to enhance their capabilities.

Following the significant advancements in NLP, various attention-based SR models have been proposed Zhou et al. (2018; 2019); Kang & McAuley (2018); Chen et al. (2019); Zhou et al. (2020); Sun et al. (2019), which have become a critical component of modern commercial recommendation systems. Attention-based SR models can be categorized into two categories: target attention based models Zhou et al. (2018; 2019) and self-attention based models Kang & McAuley (2018); Chen et al. (2019); Zhou et al. (2020); Sun et al. (2019); Liu et al. (2021). Self-attention based models need to storage Key-Value states (KV cache) of every user to save inference computational cost, thus these models need huge KV cache cost.

Generative Recommendation reformulates recommendation problems as sequential transduction tasks within a generative modeling framework. Generative recommendation models can scale up billions parameters which can significantly improve recommendation performance with large-scale data. TIGER Rajput et al. (2023) first proposed the concept of "generative recommendation" for domain-level zero-shot recommendation. RPG Hou et al. (2025) extend semantic ID-based generative recommendation to solve inefficient inference, causing by resource-intensive beam search and sequential decoding. HSTU Zhai et al. (2024) is yet another attempt to adapt Transformers to generative recommendation, after the TIGER model. It uses a point-wise normalization mechanism instead of softmax normalization, making it suitable for non-stationary vocabularies in streaming settings. FUXI-$\alpha$ Ye et al. (2025a) introduces an Adaptive Multi-channel Self-attention mechanism that distinctly models temporal, positional, and semantic features.

**Filter bubble** Gao et al. (2023a;b); Wang et al. (2023) In reality, a user's context sequence usually contains lots of homogeneous items, e.g., daily necessities, while other informative items, e.g., sports goods, may sparsely appear. Self-attention mechanism treats all interactions equally in user behavior modeling, may resulting in overemphasis on specific kinds of items and insufficient focus on others. As a result, these models tend to expose popular items to users repeatedly Wang et al. (2023), limiting the diversity of recommendations and degrading the user experience. Several methods Gao et al. (2023a;b) have been proposed to alleviate the issue, but seldom work explores from the perspective of self-attention mechanism.

## A.2  POSITIONAL ENCODING IN SEQUENTIAL RECOMMENDATIONS

Position encoding (PE) emerged as an important research topic with the rise of the Transformer architecture, as the attention mechanism is *order-invariant*. However, current researches only consider the order and ignore the time interval between items. Self-attention based SR models generally employ a naïve absolute PE approach. CAPE Yuan et al. (2025) first pointed out this issue and propose a contextual-aware PE method to enhance SR models. HSTU Zhai et al. (2024) and FUXI-$\alpha$ Ye et al. (2025a) both use a log function to map time stamp to time bias as the input features. Currently, rare researches pay attention on the importance of interaction time in user context.

## A.3  NOTATION

Some important concepts along with related symbols are listed in Table 7.

Table 7: Summary of notations

| Notation | Description |
|---|---|
| $I$ | Context item |
| $n$ | Length of context |
| $d, d_c$ | Dimension of input item embedding and compressed latent state |
| $d_{int}$ | Dimension of intermediate size of feed-forward network |
| $d_u$ | The dimension of user feature vector |
| $w$ | Window size of sliding window attention |
| $L$ | Then number of network layers |
| $V$ | The number of head in each layer |
| $\mathbf{H}^{(l)}$ | Input of $l$-th layer, $\mathbb{R}^{n \times d}$, $l \in [1, L]$ |
| $\mathbf{H}_i^{(l)}$ | Input of $i$-th item of $l$-th layer, $i \in [1, n]$, $\mathbb{R}^d$ |

# B  OTHER BLOCKS OF PT-RECFORMER

## B.1  EMBEDDING LAYER

Our approach utilizes an embedding layer to map item IDs to a high-dimensional space. The embedding layer uses a learnable embedding matrix $\mathbf{E} \in \mathbb{R}^{|\mathcal{I}| \times d}$, and we obtain the initial item embeddings $\mathbf{H}^{(0)} \in \mathbb{R}^{n \times d}$ by looking up the embedding layer. To enhance robustness and prevent overfitting, we incorporate both embedding dropout and RMSNorm after retrieving the embeddings:

$$[\mathbf{h}_1^{(1)}, ..., \mathbf{h}_n^{(1)}] = \mathbf{H}^{(1)} = \text{RMSNorm}(\text{Dropout}(\mathbf{H}^{(0)})) \tag{21}$$

## B.2  FEED-FORWARD NETWORK (FFN)

We employ a position-wise Gated Linear Units Shazeer (2020) with SiLU activation as the feed-forward network in the PT-Recformer layer.

$$\tilde{\mathbf{Y}} = \text{RMSNorm}([\mathbf{y}_1^w, ..., \mathbf{y}_n^w]) \tag{22}$$

$$\text{FFN}(\tilde{\mathbf{Y}}) = \mathbf{W}^{down}(\text{SiLU}(\mathbf{W}^{gate}\tilde{\mathbf{Y}}) \otimes \mathbf{W}^{up}\tilde{\mathbf{Y}}) \tag{23}$$

where $\mathbf{y}_j^w \in \mathbb{R}^d$ is the output of the PMLA, $\mathbf{W}^{gate}, \mathbf{W}^{up} \in \mathbb{R}^{d_{int} \times d}$, $\mathbf{W}^{down} \in \mathbb{R}^{d \times d_{int}}$. $d_{int}$ is the intermediate size of FFN. Finally, the output of a PT-Recformer layer is given by:

$$\mathbf{H} = \text{FFN}(\text{RMSNorm}([\mathbf{y}_1^w, ..., \mathbf{y}_n^w])) + [\mathbf{y}_1^w, ..., \mathbf{y}_n^w], \ \mathbf{H} \in \mathbb{R}^{n \times d} \tag{24}$$

## B.3  PREDICTION LAYER

PT-Recformer follows the prediction method of autogressive generative recommendation:

$$\hat{y}_{n+1} = \text{Softmax}(\mathbf{h}_n^{(L)}\mathbf{E}^\top) \tag{25}$$

where $\mathbf{h}_n^{(L)} \in \mathbb{R}^d$ is the last hidden representation of the stacked PT-Recformer layers' output sequence, and $\mathbf{E} \in \mathbb{R}^{|\mathcal{I}| \times d}$ denotes the representation of all candidate items.

# C  EXPERIMENTAL DETAILS

**Implementation Details.** We implement PT-Recformer and all baselines using the code provided by HLLM Chen et al. (2024) and Mamba Dao & Gu (2024). We use the Hashing Trick Rajput et al. (2023) to map the raw user ID to one of the user ID tokens, and the user ID token amount is searched in [0, 2000, 4000, 6000]. The embedding size of item and user token is 512. Then rank of compression dimension $d_c$ of PMLA is grid searched in [32, 64, 128, 256, 512]. The number of negative samples for each positive sample is set to 50. We use the Adam optimizer, whose learning rate is grid searched in [1e-4, 2e-4, 1e-3]. The maximum training epoch is 200. The batch size is set to 128 in experiments on Books and Movies&TV and 32 in experiments on Movielens-20M.

Temperature parameter $\tau$ is set to 0.05 for all experiments. Early stopping strategy is adopted to prevent overfitting with a patience of 3 epochs. All models use 8 layers and 8 head, and head dim is 64. All experiments are conducted on 8 GPUs with 32G memory. Time interval resolution factor $\beta$ is set 6.7, while $max_r tb$ is set to 4 times of maximum context length. The scaling factor of gate vector $\gamma$ is searched in [1, 2, 4]

Table 8: Statics of Datasets

| Dataset | #User | #Item | #Interaction |
|---|---|---|---|
| Books | 948,978 | 966,607 | 11,544,935 |
| Movies & TV | 691,621 | 285,252 | 7,905,714 |
| MovieLens-20M | 138,287 | 20,720 | 9,995,410 |

**Dataset**. We evaluate PT-Recformer on two large-scale datasets from Amazon Reviews 2023 Hou et al. (2024)[2]: *Books* and *Movies & TV*. We also conduct experiments on another benchmark dataset, *MovieLens-20M*[3]. We follow the protocol of HLLM Chen et al. (2024). Specifically, we retain only items and the users with at least 5 presences, and we only keep item_id, user_id and timestamp and ignore other features. The maximum length of context is set to 50 in *Books* and *Movies & TV*, and 200 in *MovieLens-20M*. Detailed datasets statistics after preprocessing is presented in Table 8.

**Metrics.** We follow the traditional sequential recommendation settings as described in the literature Wang et al. (2023). We utilize a leave-one-out approach to split the data into training, validation, and testing sets. Performance is measured using Recall@K (R@K) and NDCG@K (N@K). To account for variability, each experiment is repeated 5 times with different random seeds, and we report the average results.

## C.1 BASELINES

To fair comparison, all baselines use 8 layers with 8 heads, and the head dim is 64.

1. **SASRec** Kang & McAuley (2018) is a self-attention based sequential recommendation model, which uses the multi-head attention mechanism to recommend the next item. SASRec is a variant of Bert, so it uses post layer norm and two FC layer as feed-forward network. Additionally, SASRec assigns a learnable vector to each position and adds it to the input embeddings.

2. **SASRecV2** We implemented this baseline by changing some component of SASRec, and name it SASRecV2. We replace the post-norm in SASRec with pre-norm using RMSNorm. We employ a position-wise Gated Linear Units Shazeer (2020) with SiLU activation as the feed-forward network.

3. **MLARec** We implemented this baseline too, by removing the IGNU and SWA in PT-Recformer. Thus, we call this baseline MLARec. The target of this baseline is evaluating attention mechanism.

4. **HSTU** Zhai et al. (2024) uses a point-wise normalization mechanism instead of softmax normalization, making it suitable for non-stationary vocabularies in streaming settings. We used the best HSTU configuration proposed by FUXI-$\alpha$[4] in all experiments.

5. **FUXI-$\alpha$** Ye et al. (2025a) extended HSTU by introducing an Adaptive Multi-channel Self-attention mechanism that distinctly models temporal, positional, and semantic features. We used the best configuration proposed by itself too in experiments.

6. **Hydra** Yuan (2025) is a Mamba-based sequential recommendation models. It employs multiple low-dimensional Mamba layers and fully connected layers coupled with positional encoding to simultaneously capture historical and item information within each latent subspace. We use the best configuration reported by the paper: the expand factor of Mamba is set to 2, and dimension of state is set to 128.

---

[2] https://amazon-reviews-2023.github.io

[3] https://grouplens.org/datasets/movielens/20m/

[4] https://github.com/USTC-StarTeam/FuXi-alpha

Table 9: Statics of Time Interval

| Dataset | Mean | Std | Gini | Q3-Q1 | Max | Min |
|---------|------|-----|------|-------|-----|-----|
| Books | 7169083s | 22758627.917 | 0.8543 | 45 days | 7 years | $\leq$ 1 second |
| Movies & TV | 11148734s | 28466887.097 | 0.8209 | 100 days | 14 years | $\leq$ 1 second |
| Movielens-20M | 232498s | 3050554.664 | 0.9865 | 61 seconds | 14.7 years | $\leq$ 1 second |

## D  TIME INTERVAL OF ADJACENT ITEMS IN USER CONTEXT

Figure 4 and Table 9 both show the large-range time interval between adjacent items. The top 25% adjacent items with the maximum time interval in Amazon Books datasets is higher than 45 days, and the maximum result is 7 years. There are still 25% adjacent items whose time interval is less than 1 second. Meanwhile, the top 25% adjacent items with the maximum time interval in Movies&TV datasets is higher than 100 days, and the maximum result is 14 years. There are still 25% adjacent items whose time interval is less than 1 second. In MovieLens-20M, the top 25% adjacent items with the maximum time interval is higher than 61 seconds, and the maximum result is 14.7 years. The huge differences of time distributions between datasets also demonstrate the fact diversity of recommendation scenarios.

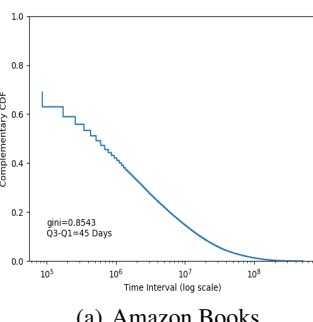 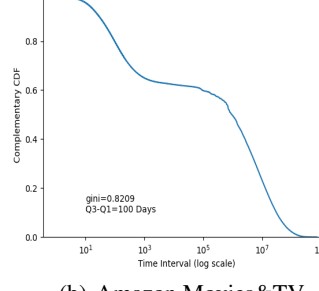 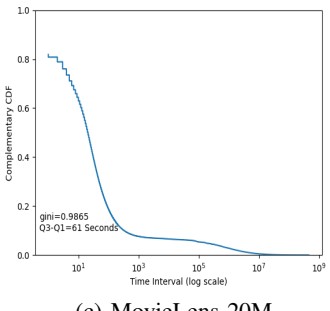

(a) Amazon Books  (b) Amazon Movies&TV  (c) MovieLens-20M

Figure 4: Complementary Cumulative distribution function of time interval of adjacent items of different datasets. The x axis is log-scale time interval between adjacent items in user context

## E  EXPERIMENT ON INDUSTRIAL DATASET

The primary objective of the online experiments is to thoroughly evaluate the effectiveness of PT-Recformer in a real-world applications, whose recommendation models is highly optimized for years.

**Industrial dataset**. The offline experiments in this study are conducted on an 15-day period large-scale dataset with several sequential features, which is sampled an collected from an online app recommendation platform. This dataset encompasses user interactions across five key tasks, including app download, activation, and other critical in-app behaviors, providing a comprehensive reflection of real-world user engagement patterns. To ensure the experimental setup closely mirrors practical recommendation scenarios, the dataset is partitioned in chronological order into three distinct subsets: a training set (80%), a validation set (10%), and a testing set (10%). This temporal splitting strategy helps prevent data leakage and ensures that the model evaluation reflects realistic deployment conditions.

**Experimental Setups**. For performance evaluation, we adopt the Area Under the Curve (AUC) metric, which is widely recognized for measuring the ranking quality of recommendation systems. This choice aligns with the evaluation protocols used in prior studies on public datasets, enabling fair comparisons with existing methods. The baseline model selected for comparison is a highly competitive multi-task learning (MTL) framework, which incorporates multiple advanced modules from state-of-the-art MTL approaches in the literature. This baseline has been rigorously optimized to

deliver strong performance across various tasks, making it a challenging benchmark for our proposed method.

### E.1 Offline Performance

As shown in Table 10, the experimental results highlight the superior performance of our PT-Recformer method compared to the baseline across all five tasks in the product dataset. The consistent improvements demonstrate the effectiveness of our approach in handling complex multi-task learning scenarios, particularly in industrial-scale applications where robustness and generalization are critical. Further analysis reveals that PT-Recformer's gains are statistically significant, reinforcing its potential for real-world deployment in large-scale recommendation systems.

Table 10: We evaluate our approach using an industrial dataset, with the AUC as the primary performance metric.

| Method | Task 1 | Task 2 | Task 3 |
|---|---|---|---|
| baseline | 0.978873 | 0.854974 | 0.877551 |
| PT-Recformer | 0.979986 | 0.85661 | 0.878742 |
| Method | Task 4 | Task 5 | Task 6 |
| baseline | 0.837706 | 0.850326 | 0.800892 |
| PT-Recformer | 0.838862 | 0.851459 | 0.801975 |

## F Ablation Study

### F.1 Window Size

We evaluate the impact of window size of SWA in the following experiments. All models use 8 layers with 8 heads. As shown in Table 11, when window size is half of maximum context length, models achieve best performance. Actually, when window size is the quarter of maximum context length (12 in *Books* and *25* in *Movielens-20M*), PT-Recformer is able to outperforms SOTA baselines. Adopting SWA and low rank compression at same time, PT-Recformer can outperform SOTA baselines with only 12% memory of SASRec for KV cache.

Table 11: Evaluation of different window size of SWA. $d_c$ is 256. The results are average of 5 times training with different random seeds. The best results on the same dataset are bold with $p < 0.05$.

| Dataset | $w$ | Metrics | | | | | |
|---|---|---|---|---|---|---|---|
| | | R@10 | R@50 | R@200 | N@10 | N@50 | N@200 |
| Books | 6 | 7.45348 | 13.69685 | 22.05665 | 4.70838 | 6.06761 | 7.31844 |
| | 12 | 7.49363 | 13.75571 | 22.10471 | 4.73711 | 6.10066 | 7.35050 |
| | 25 | **7.56688** | **13.95876** | **22.43541** | **4.76576** | **6.15792** | **7.42781** |
| | 50 | 7.52860 | 13.89112 | 22.31914 | 4.75319 | 6.13932 | 7.40226 |
| Movielens-20M | 12 | 14.72728 | 28.90077 | 45.54704 | 8.74585 | 11.83421 | 14.34211 |
| | 25 | 16.87166 | 31.45727 | 47.99408 | 10.04965 | 13.24414 | 15.72477 |
| | 50 | 17.00045 | 31.72130 | 48.58008 | 10.38008 | 13.59403 | 16.13114 |
| | 100 | **17.62509** | **32.19138** | **48.52212** | **10.61433** | **13.80393** | **16.25550** |
| | 200 | 17.58645 | 31.65046 | 48.16150 | 10.44333 | 13.60248 | 16.18744 |

### F.2 Positional QK bias

We compare our positional QK bias method with two tricks on our PT-Recformer, *without bias* and *all bias*. Without bias means remove all bias of linear layer in multi-head self-attention like Llama. All bias is opposite, which means all linear layer adopts bias. As shown in Table 12, our position QK bias improves PT-Recformer on both datasets and achieves best performance. Meanwhile, all bias trick deteriorates backbone and achieves worst results. The improvement on Movielens-20M is higher than on Books. This is because the user context in Movielnes-20M is longer than in Books.

Table 12: Evaluation on positional QK bias. Then window size is 12 on Books and 50 on Movielens-20M, respectively. $d_c$ is 256. The best results on the same dataset are bold with $p < 0.05$.

| Dataset | Method | Metrics | | | | | |
|---|---|---|---|---|---|---|---|
| | | R@10 | R@50 | R@200 | N@10 | N@50 | N@200 |
| Books | w/o bias | 7.45852 | 13.64116 | 21.90497 | **4.74645** | 6.09320 | 7.33051 |
| | all bias | 7.40268 | 13.49063 | 21.72955 | 4.70031 | 6.02322 | 7.25741 |
| | positional QK bias | **7.49363** | **13.75571** | **22.10471** | 4.73711 | **6.10066** | **7.35050** |
| Movielens-20M | w/o bias | 16.72999 | 30.52354 | 45.96561 | 10.13695 | 13.14934 | 15.47478 |
| | all bias | 16.45953 | 30.28527 | 46.92511 | 9.75666 | 12.77943 | 15.27802 |
| | positional QK bias | **17.00045** | **31.72130** | **48.58008** | **10.38008** | **13.59403** | **16.13114** |

## F.3 DIFFERENT INTERACTION MODULE

We compare our IGNU with Gate Neural Unit (Gate NU) Chang et al. (2023) to reveal its contribution. The experimental results are demonstrated in Table 13. PT-Recformer with our proposed IGNU outperform Gate NU at all metrics across three benchmark datasets. These results show the great design of our IGNU.

Table 13: Evaluation on different Interaction Module. Recall and NDCG values are averages over 5 random seeds and only keep 5 decimal. The best results on the same dataset are bold with $p < 0.05$. The $d_c$ is 256, and the window size $w$ is 25 and 100 on Books and Movielens-20M, respectively.

| Dataset | Method | R@10 | R@50 | R@200 | N@10 | N@50 | N@200 |
|---|---|---|---|---|---|---|---|
| Books | None | 7.39836 | 13.35622 | 21.29337 | 4.74132 | 6.03856 | 7.22654 |
| | Gate NU | 7.49305 | 13.82464 | 22.27121 | 4.73221 | 6.1101 | 7.37492 |
| | IGNU | **7.56688** | **13.95876** | **22.43541** | **4.76576** | **6.15792** | **7.42781** |
| Movies&TV | None | 9.19535 | 17.40621 | 28.11699 | 5.65884 | 7.44542 | 9.05092 |
| | Gate NU | 9.23613 | 17.12267 | 27.34344 | 5.78309 | 7.50013 | 9.03239 |
| | IGNU | **9.23772** | **17.51174** | **28.50900** | **5.79187** | **7.52726** | **9.17153** |
| MovieLens-20M | None | 16.65271 | 31.06446 | 47.67210 | 9.98470 | 13.14817 | 15.65912 |
| | Gate NU | 17.16788 | 31.61826 | 48.69599 | 10.1867 | 13.35203 | 15.91724 |
| | IGNU | **17.62509** | **32.19138** | **48.52212** | **10.61433** | **13.80393** | **16.25550** |

## F.4 IMPACT OF USER PROFILE FEATURE

We follow TIGER Rajput et al. (2023), using the Hashing Trick to map the raw user ID to one of the 2000 user ID tokens, addressing optimization challenges for raw ID embeddings. We evaluated our model with different user token counts to investigate noise robustness, with results presented below:

As shown in the Table 14, performance begins to decline once the user token count exceeds 2000, indicating that user profile attribute introduces noise. Nevertheless, even at 6000 tokens all metrics remain above those of MLARec (no user profile attribute), demonstrating our model's robustness to such noise and underscoring the strength of its design.

Table 14: Evaluation on different count of user tokens on Amazon Books. Recall and NDCG values are averages over 5 random seeds and only keep 5 decimal. The best results are bold with $p < 0.05$. The $d_c$ is 256, and the window size $w$ is 25 and 100 on Books and Movielens-20M, respectively.

| User Token Count | R@10 | R@50 | R@200 | N@10 | N@50 | N@200 |
|---|---|---|---|---|---|---|
| 0 | 7.39836 | 13.35622 | 21.29337 | 4.74132 | 6.03856 | 7.22654 |
| 2000 | **7.56688** | **13.95876** | **22.43541** | **4.76576** | **6.15792** | **7.42781** |
| 4000 | 7.48974 | 13.83802 | 22.29251 | 4.72678 | 6.10882 | 7.37566 |
| 6000 | 7.38239 | 13.55971 | 21.91246 | 4.67387 | 6.01724 | 7.26752 |

## F.5 IMPACT OF SCALING FACTOR $\gamma$

In this experiment, we aim to evaluate the impact of scaling factor $\gamma$ of IGNU on Amazon Books dataset. The $\gamma$ is searched within [1, 2, 4]. Window size is 25, and rank size is 2256. The experimental result is demonstrated in Table 15. When $\gamma = 1$, the model achieved best performance.

Table 15: Evaluation on scaling factor $\gamma$ on Amazon Books. Recall and NDCG values are averages over 5 random seeds and only keep 5 decimal. The best results are bold with $p < 0.05$. The $d_c$ is 256, and the window size $w$ is 25.

| $\gamma$ | R@10 | R@50 | R@200 | N@10 | N@50 | N@200 |
|---|---|---|---|---|---|---|
| 1 | 7.49622 | 13.61641 | 21.87576 | 4.76353 | 6.09585 | 7.33274 |
| 2 | **7.56688** | **13.95876** | **22.43541** | **4.76576** | **6.15792** | **7.42781** |
| 4 | 7.31993 | 13.36328 | 21.65530 | 4.67350 | 5.98949 | 7.23044 |

## F.6 COMPARISON WITH LONG CONTEXT MODEL

Mainstream long sequence recommendation model retains recent interactions while sampling distant ones (e.g., ETA Chen et al. (2022)) to compress context sequences. The sliding-window strategy achieves dual objectives: lower layers focus on recent behaviors, while higher layers compress long-term sequences into compact KV caches.

On MovieLens-20M, our method with a window size of 100 outperforms ETA (50 most recent items and sampling 50 remote items), demonstrating that sliding-window attention is more effective than sampling for modeling long sequences.

Table 16: Comparison with long context model on MovieLens-20M. Recall and NDCG values are averages over 5 random seeds and only keep 5 decimal. The best results are bold with $p < 0.05$. The $d_c$ is 256, and the window size $w$ is 100.

| Method | R@10 | R@50 | R@200 | N@10 | N@50 | N@200 |
|---|---|---|---|---|---|---|
| ETA | 15.87353 | 29.99549 | 47.12473 | 9.40988 | 12.48856 | 15.06689 |
| PT-Recformer | **17.62509** | **32.19138** | **48.52212** | **10.61433** | **13.80393** | **16.25550** |

