# OpenReview forum: "Personalized and Temporal-aware Attention for Efficient Generative Recommendation"
_ICLR.cc/2026/Conference — Submitted to ICLR 2026_

### Official Review · Reviewer_tKAs · 2025-10-25

**Soundness:** 2
**Presentation:** 2
**Contribution:** 3
**Rating:** 2
**Confidence:** 4

**Summary:**

The paper introduces PT-Recformer, a personalized and temporal-aware Transformer model for generative recommendation, addressing inefficiencies in self-attention-based sequential recommendation (SR) systems. It identifies two critical issues: inadequate temporal information modeling and KV cache redundancy. The proposed solution includes Rotary Temporal Encoding (RoTE) for handling wide-ranging time intervals and a Personalized Multi-head Latent Attention (PMLA) mechanism to integrate user-specific features, reducing KV cache usage by 88%. Experiments on benchmark datasets (Amazon Books, Movies & TV, MovieLens-20M) and online A/B testing show superior performance and a 3.13% eCPM increase, but the methodology and claims require rigorous scrutiny.

**Strengths:**

- **Originality**: Attempts to tailor self-attention for SR with RoTE and PMLA, though heavily derivative.
- **Quality**: Demonstrates some performance gains (e.g., 15.5% over HSTU-large on Books), but experimental design flaws temper reliability.
- **Clarity**: Figures provide visual support, though annotations are inadequate.
- **Significance**: Addresses practical efficiency concerns, but the solution’s novelty and scalability are questionable.

**Weaknesses:**

- **Methodological Flaws**: The temporal interval analysis (e.g., 45-day gaps in Amazon Books) lacks statistical significance testing, and RoTE’s logarithmic bias (Equation 2) is arbitrarily parameterized without optimization studies.
- **KV Cache Claims**: The 88% reduction is based on a fixed window size (w) without exploring dynamic adaptation, potentially sacrificing long-range context accuracy. No comparison with state-of-the-art memory optimization techniques (e.g., sparse attention) is provided.
- **A/B Testing**: The 3.13% eCPM increase lacks transparency on sample size, test duration, or user segmentation, rendering it inconclusive.
- **Reproducibility**: Hyperparameters (e.g., β, γ, w) are not systematically tuned or reported, and code availability (anonymous link) cannot be verified pre-publication.
- **Oversight**: Ignores potential biases in dataset temporal distributions and their impact on model performance.

**Questions:**

1. Can the authors provide statistical tests (e.g., t-tests) to validate the significance of temporal interval differences across datasets, and justify the choice of β in Equation 2?
2. How does PT-Recformer’s accuracy degrade with varying window sizes (w) in PMLA, and why was no dynamic window adjustment tested against sparse attention methods?
3. What was the A/B test’s sample size, duration, and control group design to support the 3.13% eCPM claim, and how were confounding factors (e.g., seasonality) mitigated?
4. Why were hyperparameters (β, γ, w, τ) not optimized via grid search or reported with sensitivity analysis, and how does this affect reproducibility?
5. Can the authors compare PT-Recformer’s KV cache efficiency with recent memory-optimized Transformers (e.g., Longformer, Performer) to substantiate the 88% reduction claim?

---

> ### Comment · Reviewer_xc4X · 2025-11-12
>
> fwiw this review seems poorly written..
>
> - "code availability (anonymous link) cannot be verified" - the authors did provide a working repo link!
> - [A/B test] missing "sample size, ..., and control group design" - applies to 95% of accepted KDD industrial track papers. I guess you *could* also critique KDD's review process?
> - "recent memory-optimized Transformers (e.g., Longformer, Performer)" (as someone who work on attention mechanisms and publishes somewhat actively, I do not believe Longformer/Performer belong in the same sentence as "memory-optimized Transformers" given it's already 2025)
>
> (from reviewer xc4X; while I do see many flaws with this paper, this review seems unfair and I hope it's not generated by LLMs)

---

> > ### Comment · Area_Chair_Hjud · 2025-11-20
> >
> > Thanks for pointing out the issue. I will check this review carefully and then decide whether to report to PC or not.
> >
> > AC

---

### Official Review · Reviewer_7C7b · 2025-10-28

**Soundness:** 3
**Presentation:** 2
**Contribution:** 2
**Rating:** 2
**Confidence:** 5

**Summary:**

This paper proposes a personalized and temporal-aware transformer for generative recommendation. They introduce a rotary temporal encoding method to model the temporal variations inherent in user contexts, and a personalized multi-head latent attention mechanism to enhance the expressive power. They demonstrated its performance using both public benchmarks and a real-world online platform and compared it with other generative recommendation methods.

**Strengths:**

1. This paper considers the temporal information of each sequence in the sequence recommendation task and innovatively proposes rotary temporal encoding to model temporal information.

2. This article found that only the closest items are focused on in self-attention from the perspective of KV cache cost, so personalized multi-head latent attention was designed.

3. The authors tested PT-Recformer's performance and compared it with previous GR methods.

**Weaknesses:**

1. While PT-Recformer claims to model temporal information, there is no actual visualization of Rotary Temporal Encoding (RoTE).

2. There are several typos throughout the manuscript; ones I noticed include:
      - γ is a scalar hyperparameter and should be expressed as “⊗” or simply “·” to indicate element-wise scaling.
      - The dimension of $b_q$ in Eq. (7) should be $b_q \in \mathbb{R} ^{n_h d^R_h}$?
      - The writing and certain claims need refinement.

3.  Many modules in the paper appear to be incrementally stacked. For example, ROTE is a temporal extension of ROPE, and sliding window attention directly inherits Longformer. The multi-head attention mechanism does not seem to have any fundamental innovations. Can it really bring diversity to recommendations?

**Questions:**

1. HSTU and other structures can unify the **Retrieval** and **Ranking** tasks. How about PT-Recformer？
2. Whether the dimension of $b_q$ in Eq. (7) should be $b_q \in \mathbb{R} ^{n_h d^R_h}$?
3. Does Personalized Multi-head Latent Attention bring diversity to recommendations? Are there any experimental results to support this?

**Details Of Ethics Concerns:**

No ethics concerns

---

### Official Review · Reviewer_iSR3 · 2025-10-28

**Soundness:** 3
**Presentation:** 3
**Contribution:** 2
**Rating:** 2
**Confidence:** 4

**Summary:**

This paper has focused on the sequential recommendation field. The authors found that inadequate modeling of temporal information and KV cache redundancy may impair the performance of the self-attention-based SRS model and lead to inefficiency challenges. To address these issues, this paper proposed a novel rotary encoding method to fully integrate temporal information and a personalized multi-head latent attention mechanism to relieve the inefficiency issue. Extensive experiments validated the effectiveness and efficiency of the proposed method.

**Strengths:**

+ S1. This paper is well-organized and -written, making it easy to follow.
+ S2. Extensive experiments have been conducted.
+ S3. The code is released, making it easy to reproduce.

**Weaknesses:**

- W1. The illustration that HSTU and Fuxi-$\alpha$ fail to fully leverage the temporal information is arbitrary. More analysis is needed to validate such a conclusion.
- W2. The term "generative recommendation" used in this paper does not seem accurate, which may confuse readers. In general, the generative recommendation refers to the model that aligns tokens of items with natural language and outputs the semantic ID of items, like TIGER [1]. The paper proposed in this paper belongs to discriminative recommendation.
- W3. Though this paper has addressed the efficiency issue in sequential recommendation, many recent efficient sequential recommendation baselines have been ignored, such as.
- W4. The experiments on training and inference time should include the baseline model Hydra, because it often achieves comparable performance.
- W5. I found the training objective of this paper is InfoNCE, which is different from the objective of Fuxi-$\alpha$ and HSTU. Different training objectives may lead to unfair comparison.



[1].Rajput, Shashank, et al. "Recommender systems with generative retrieval." *Advances in Neural Information Processing Systems* 36 (2023): 10299-10315.



[2]. Li, Muyang, et al. "MLP4Rec: A pure MLP architecture for sequential recommendations." *arXiv preprint [arXiv:2204.11510](https://arxiv.org/abs/2204.11510)* (2022).

**Questions:**

All my questions have been included in the weakness section.

---

### Official Review · Reviewer_xc4X · 2025-11-09

**Soundness:** 3
**Presentation:** 2
**Contribution:** 3
**Rating:** 6
**Confidence:** 4

**Summary:**

This paper proposes PT-Recformer, a novel transformer-like architecture for sequential recommendation that addresses two inefficiencies in current self-attention based models: inadequate temporal modeling and KV cache redundancy. The authors introduce Rotary Temporal Encoding (RoTE) to handle temporal variations in user interaction sequences (ranging from seconds to years), and Personalized Multi-head Latent Attention (PMLA) that integrates user profile features while reducing KV cache costs through low-rank compression and sliding window attention. The method is evaluated on three public benchmarks and deployed in a commercial system, showing improvements over SOTA baselines with as little as 12% of the KV cache storage and 75% training time of vanilla self-attention models.

**Strengths:**

S1: Problem motivation: The paper provides empirical evidence (Figure 1) showing that self-attention models focus predominantly on the last few items, leading to massive KV cache redundancy.

S2. Technical contributions: The paper makes the following technical contributions, including a/ RoTE with learnable frequencies, an elegant solution for handling extreme temporal ranges in recommendation data, b/ Integration of personalized features directly into attention mechanisms (rather than late fusion), and c/ Combination of low-rank compression with sliding window attention to reduce KV cache sizes

S3. Comprehensive experimental validations: a/ The authors conduct extensive experiments on three public benchmarks with consistent improvements, with thorough ablation studies (eg Table 4). b/  The authors deployed the solution in real-world settings, with A/B testing showing 3.13% eCPM improvement.

S4. Overall, the method seems like a strong improvement over current SotA, with as little as 12% KV cache, 75% training time, and faster inference, which is significant for practical deployment.

**Weaknesses:**

W1: Limited novelty in individual components: While the combination is novel, individual components (sliding window attention, low-rank compression, RoPE) are well-established techniques. The primary contribution of this paper appears to be in their adaptation and combination for recommendation.

W2. Design
* Despite observation of "self-attention models focusing predominantly on the last few items", the optimal hyperparameter for ML-20M is 100, suggesting that sliding window may not be that effective.
* While PMLA is a step forward on top of MLA, the usage of user profile tokens can introduce leakage.
* The temporal relative attention bias mechanism in HSTU can be combined with PMLA too. What are the tradeoffs of (temporal aware) RAB vs RoPE on top of PMLA?

W3. Experiment comparisons
* Table 1 comparison is misleading; we should not compare sliding window style attention and full attention in the same table, as one can easily integrate sliding window attention into various baselines (eg SASRec, HSTU, etc).
* Table 3: 0.37B model being almost as fast as 1B HSTU models suggest PT-Recformer implementation may not be sufficiently optimized?
* While the solution is deployed in production, additional details for baseline/test would be helpful - eg scale of baseline model, distribution of sequence lengths, model refresh frequencies, whether dataset consists of news-like items with shorter timespans or retail with seasonal patterns, etc.

W4. Writing
- Incorrect citations. https://arxiv.org/abs/2111.11294 was the first work to study scaling laws in recommendations. Zhang et al. (2023)'s work came much later.
- Figure 1 and Figure 2(b) are hard to read
- Table 2 suggests that the results are accurate up to .00001 Recall@K but I doubt this is the case. Why not report variance given you've already conducted multiple runs?
- Mathematical notation could be cleaner
- Typos: Time interval resolution factor β is set 6.7 -- ??
- Formatting in various Tables could be improved (eg 7169083s in Table 9 should be converted to hours)

**Questions:**

Please see above. Addressing issues in W2/W3/W4 would be especially helpful.

---

### Meta-Review · Area_Chair_v7rJ · 2026-01-01

**Summary:**

This paper proposes PT-Recformer, a personalized and temporal-aware transformer architecture for sequential recommendation, aiming to improve temporal modeling and reduce KV cache redundancy. Reviewers acknowledged the practical motivation and the reported efficiency gains, including reduced KV cache usage and positive online A/B testing results. However, concerns were raised regarding the novelty of the proposed components, experimental rigor, clarity of presentation, and the strength of the empirical evidence supporting the claims.

**Reviewer Concerns:**

The main concerns center on the limited novelty of individual components, which largely build upon existing techniques such as RoPE-style encodings, sliding-window attention, and low-rank compression, as well as potentially misleading or incomplete experimental comparisons. Reviewers questioned the fairness of baseline selection, the interpretation of temporal modeling benefits, and the lack of robustness analysis and variance reporting. Additional issues were noted in presentation quality, terminology, citation accuracy, and insufficient transparency in the reported online A/B testing. While the rebuttal addressed some clarifications, these core concerns were not fully resolved.

**Reviewer Scores:**

The reviewer scores were mixed but leaned toward rejection, with multiple reviewers recommending rejection and expressing high confidence. Although one reviewer rated the paper marginally above the acceptance threshold, there was no strong consensus that the work meets the novelty, clarity, and experimental standards expected for acceptance at ICLR.

---

### Decision · Program_Chairs · 2026-01-26

Reject